# Evaluation of Late Postural Complications in Breast Cancer Patients Undergoing Breast-Conserving Therapy in Relation to the Type of Axillary Intervention-Cross-Sectional Study

**DOI:** 10.3390/jcm10071432

**Published:** 2021-04-01

**Authors:** Iwona Głowacka-Mrotek, Magdalena Tarkowska, Lukasz Leksowski, Tomasz Nowikiewicz, Wojciech Zegarski

**Affiliations:** 1Department of Rehabilitation, Nicolaus Copernicus University in Toruń, Collegium Medicum in Bydgoszcz, 85-094 Bydgoszcz, Poland; lukasz.leksowski@cm.umk.pl; 2Department of Physiotherapy, Nicolaus Copernicus University in Toruń, Collegium Medicum in Bydgoszcz, 85-094 Bydgoszcz, Poland; magdalena.sowa@cm.umk.pl; 3Department of Surgical Oncology, Nicolaus Copernicus University in Toruń, Collegium Medicum in Bydgoszcz, 85-067 Bydgoszcz, Poland; tomasz.nowikiewicz@cm.umk.pl (T.N.); zegarskiw@cm.umk.pl (W.Z.); 4Department of Clinical Breast Cancer and Reconstructive Surgery, Prof. Franciszek Łukaszczyk Oncology Center, Romanowskiej Street, 85-796 Bydgoszcz, Poland

**Keywords:** breast-conservingtherapy, body posture, axillary lymph node dissection, sentinel lymph node dissection

## Abstract

Purpose: The aim of the study was to evaluate posture in patients undergoing breast-conserving therapy (BCT) in relation to the type of surgical intervention to the axilla. Methods: The study was conducted on patients who had undergone breast-conserving surgical treatment for breast cancer 5–6 years earlier. In 54 patients, BCT+ALND (axillary lymph node dissection) was performed, while 63 patients were subjected to BCT+SLND (sentinel lymph node dissection). The control group consisted of 54 females. The study was conducted using digital postural assessment. Results: No statistically significant differences were observed with respect to the parameters between the BCT+SLNB and BCT+ALND groups (*p* > 0.05). However, the differences were highly significant between the CG (control group) and the studied groups (BCT+ALND, BCT+SLNB) for the following parameters: BETA angle of thoracolumbar spine inclination (*p* = 0.002), GAMMA angle of thoracic spine inclination (*p* = 0.0044), TKA (thoracic kyphosis angle) (*p* < 0.0001) and shoulder level inclination (*p* = 0.0004). The BCT+ALND patients were characterized by higher dependency of raised shoulder (*p* = 0.0028) and inferior angle of the scapula (*p* = 0.00018) on the operated side compared to BCT+SLNB patients. Conclusions: Postural imbalance occurs independent of the type of axillary intervention. Disturbances within the upper torso (abnormal position of shoulders and inferior angles of scapulae) are more pronounced in patients after ALND.

## 1. Introduction

In oncology, efforts are made to minimize adverse sequelae while maintaining oncological radicality, and in spite of that, late complications are still observed in breast cancer patients subjected to breast-conserving therapy [1,2]. The type of surgical intervention to the axilla also plays an important role. Depending on the initial staging of the disease, sentinel lymph node biopsy (SLNB) or axillary lymph node dissection (ALND) may be performed. Randomized clinical trials and meta-analyses showed significantly higher rate of late complications in the case of ALND [3,4]. At the same time, it was established that BCT+SLNB is not free of adverse sequelae. In patients undergoing BCT+SLNB, a limited motion range of the shoulder, lymphedema and hypoesthesia are observed [5].

In the literature, there are studies evaluating posture; however, most of them refer to breast cancer patients undergoing mastectomy. Research shows that this type of intervention is associated with deepened thoracic kyphosis, flattened lumbar lordosis, unequal shoulder level on both sides, and foot deformation [6,7,8]. Other studies suggest a beneficial effect of concurrent breast reconstruction on posture [9,10]. There are only single reports on the effect of breast-conserving therapy on posture in breast cancer patients [11,12]. Previous studies have demonstrated that, when compared to patients treated with BCT approach, women who underwent mastectomy were more often observed to present the following posture abnormalities: greater trunk inclination angle, more prominent scapular and shoulder asymmetry, greater forward trunk inclination, more prominent scapular asymmetry, greater pelvic tilt angle, significant disproportion in the prominence of both inferior scapular angles. Previous research in this field has also confirmed that problems in the arm and shoulder, including lymphedema, have a significantly higher incidence after mastectomy, in comparison to BCT [13].

There is a lack of evidence regarding the effect of axillary intervention (ALND or SLND) on patient’s posture.

The aim of the study was to evaluate the long-term effect of treatment on posture in breast cancer patients receiving breast-conserving therapy depending on the type of axillary intervention.

## 2. Materials and Methods

### 2.1. Study Design

A cross-sectional study was conducted between January 2018 and March 2019 on patients who had undergone breast-conserving surgical treatment for breast cancer 5–6 years earlier, based on the acceptance of the Bioethics Committee of Collegium Medicum in Bydgoszcz (KB 8/2018). Informed consent was obtained from all individual participants included in the study. The study design was as follows: medical records were subjected to preliminary analysis against the inclusion and exclusion criteria, then the patients undergoing BCT surgery between January 2012 and December 2012 were contacted by phone in order to invite them for a voluntary and free-of-charge postural assessment. Women from the control group were recruited from local senior clubs. Study protocol was developed by researchers. The study variables such as age, disease stage, type of adjuvant treatment, disease stage, number of lymph nodes removed were collected from the patient’s medical record. During the process of qualification for adjuvant therapy, treatment was provided according to generally accepted guidelines regarding breast cancer [14,15]. Radical 3D radiotherapy was planned and administered to all patients who underwent BCT (radiotherapy with 6 MeV X-rays was applied to the entire breast with margin and adjusted to anatomical structures—a total dose of 50 Gy was administered in 25 fractions). An additional 10 Gy of radiation (boost dose) were applied to the primary tumor bed, including a 1–2 cm tissue margin.

If metastatic changes were identified in at least 4 axillary lymph nodes (or an extra capsular infiltration of metastatic changes was present—irrespective of the number of involved lymph nodes), the area treated with radiation also included lymph node regions (axilla region and medial portion of the supraclavicular region).

We decided to examine women 5 years after surgery to see the long-time effect on posture.

The inclusion criteria contained:-informed consent for participation in the study;-5 years since surgery.

The exclusion criteria included:-patients with conversion to breast amputation or an extended intervention to the axilla;-diagnosed with: neurological (Parkinson’s disease, peripheral neuropathy), musculoskeletal (inflammation, scoliosis) or rheumatoid disorders;-with a history of trauma prior to or during the course of study;-bilateral intervention;-patients diagnosed with cancer metastases over the course of the study;-undergoing breast reconstruction within the studied period;-patients with lymphoedema of the limb on the operated side.

The physical examination was carried out according to the protocol below:Height measurementWeight measurement on clinical scales without shoes

Based on the height and weight, the body mass index (BMI) was calculated for each patient.
3.Photogrammetric evaluation, i.e., digital evaluation of posture (DEP). The physical basis of this method is the Moire phenomenon [16]. The evaluation by the Moire method was conducted in the following manner:
-specific anthropometric points were marked on the patient’s back (cervical spinal processes, posterior superior iliac spines, inferior angles of scapulae). The patient would stay freely in an upright position, feet apart, arms resting along the waist, head pointing forward. On the screen, the patient’s back down to the intergluteal cleft was displayed, bra having been removed and underwear not pressing buttocks.-a preview was launched, the room was darkened and the lighting was turned on. The device was placed at the appropriate height so that the center of the back was displayed at the center of the screen. The camera obtained a series of pictures. From all the photographs, the one with optimal position of the pelvis was chosen.-the selected picture was evaluated using digital software. The measured values were collected in an Excel spreadsheet.

Photogrammetric study enables to evaluate about 60 postural parameters. For statistical analysis, the parameters describing posture in sagittal and coronary planes were used.

Sagittal plane parameters:
-ALPHA–lumbosacral spine angle-BETA–thoracolumbar spine angle-GAMMA–thoracic spine angle-LLA–lumbar lordosis angle-TKA–thoracic kyphosis angle-TBA torso bend angle. This parameter describes anterior or posterior bend of the torso based on the analysis of the line connecting spinal processes C7 to S1. Its value is positive for posterior bend and negative for anterior bend.

Coronary plane parameters (TTA, PTA, SLA, SL, SP, VP)
-TTA–torso tilt angle. It describes lateral tilt of the torso to the left or right based on the analysis of the line connecting spinal processes C7 to S1. Its value is positive for right tilt and negative for left tilt.-PTA–pelvic tilt angle. It refers to the position of the posterior superior iliac spines. Positive values refer to right spina above the left spina, while negative values describe the opposite situation. Its numerical value corresponds with the difference in height between both iliac spines expressed in millimeters.-SLA–shoulder level angle. It describes left and right shoulder. Its value is expressed in degrees; positive values describe the situation when the right shoulder is above the left shoulder, while negative value denotes left shoulder being higher. The value in millimeters expresses the difference in shoulder level on both sides.-SL–scapula level. It describes the difference in inferior angle level of the scapulae, expressed in degrees; if the left scapula is raised, the value is negative, otherwise it is positive.-SP–scapula protrusion. It characterizes shoulder blade protrusion from the spine in millimeters.-VP–vertebra position. It describes the displacement of spinal processes from the midline in millimeters; positive for right displacement, negative for left displacement.

### 2.2. Statistical Analysis

Statistical analysis was performed using PQStat statistical package, version v. 1.6.4.122 (PQStat, Poznań, Poland). Age, height, weight and difference between groups were analyzed using Kruskal–Wallis and post hoc Dunn’s tests. The advancement of the disease and type of adjuvant therapy in different groups was analyzed using the Chi-squared test. The number of removed lymph nodes and metastatic lymph nodes were analyzed using Mann–Whitney U test. The postural parameters irrespective of the operated side were analyzed using Kruskal–Wallis and post hoc Dunn’s tests. The postural parameters dependent on the operated side were compared using the Chi-squared test. The correlation between the operated side and body posture was investigated with Mental–Haenszel stratified analysis. The probability level of *p* < 0.05 was considered significant, while the level of *p* < 0.01 was assumed to be highly significant.

## 3. Results

### 3.1. Group Characteristics

A total of 472 breast-conserving surgeries were performed in the Department of Breast Cancer and Reconstructive Surgery, Oncology Center in Bydgoszcz between January 2012 and December 2012. We have made an attempt to establish contact with 472 patients (one patient remained anonymous). The inclusion criteria were met by 117 females, 54 with BCT+ALND and 63 with BCT+SLNB. The control group consisted of 54 women of similar age, who agreed to participate in the study. The studied group consisted of 117 women, 63 of whom underwent BCT+SLNB, while 54 women underwent BCT+ALND. The control group consisted of 54 individuals. Scheme describing patient exclusion from the study are shown in Figure 1.

No statistically significant differences were found between the groups with regard to body weight, height, operated side. Statistically significant differences were noted as to the staging of the disease, number of resected lymph nodes, number of the involved lymph nodes and type of adjuvant treatment.

Detailed clinical and socio-demographics data are shown in Table 1.

### 3.2. Analysis of Postural Parameters in Studied Groups (BCT+ALND, BCT+SLNB, CG).

The following parameters were compared: ALPHA, BETA, GAMMA, TKA, LLA, PTA (mm), SLA (mm), SP (mm). No statistically significant differences between the BCT+SLNB and BCT+ALND groups were found (*p* > 0.05). However, between the control and studied groups (BCT-SLND and BCT+ALND), highly statistically significant differences were observed with respect to the following parameters:-difference in BETA between the control and BCT+SLNB group was statistically significant (*p* = 0.002);-difference in GAMMA between the control and BCT+ALND group was highly statistically significant (*p* = 0.0044);-difference in TKA between CG vs. BCT+ALND, and between CG vs. BCT+ALND was highly statistically significant (*p* < 0.0001);-difference in SLA between CG vs. BCT+ALND, and between CG vs. BCT+ALND was highly statistically significant (*p* = 0.0001) (Table 2).

### 3.3. Analysis of Correlations between Groups (BCT+ALND, BCT+SLNB) and Parameters (TBA, TTA, PTA, SLA, SL, VP) in Relation to the Operated Side (Left or Right).

Next, the correlation between the operated side and the change in selected postural parameters (TBA, TTA, PTA, SLA, SL, VP) was evaluated. For the sake of the analysis, patients from the BCT+SLND and BCT+ALND groups were merged into one group. Then, the parameters were analyzed according to the operated side (left or right). No statistically significant differences in TBA and PTA were found (*p* > 0.05). However, statistically significant difference was observed for TTA (*p* = 0.0264). Highly statistically significant differences were observed for the following parameters: SLA (*p* < 0.0001), SL (*p* = 0.0010), and SP (*p* < 0.0001), indicating that those parameters depend on side of surgery (Table 3).

Postural parameters dependent on the operated side (TBA, TTA, PTA, SLA, UL, UK) were also evaluated with respect to the BCT+ALND and BCT+SLNB groups. Highly statistically significant differences in SLA (*p* = 0.0028) and SL (*p* = 0.0018) were observed depending on the operated side. The patients from the BCT+ALND group were characterized by a raised shoulder position and lowered angle of the scapula on the operated breast side compared to patients from the BCT+SLNB group (Table 4).

## 4. Discussion

In this study, we examined the relationship between postural imbalance in breast cancer patients undergoing BCT depending on type of axillary intervention (ALND vs. SLNB). Our study is the first observation study evaluating long-term effects on posture in breast cancer patients depending on type of axillary intervention. The validity of the study is raised by inclusion of control group. The increased life expectancy of women diagnosed with breast cancer suggests that many of these women may be living with the sequelae of treatment [17], which is why we decided to check the long-term effect on posture among women operated with BCT.

We examined our groups with the photogrammetric method. In our study, we used photogrammetric evaluation with Moire phenomenon. This method is based on Moire conturography observed in the optics. When a light ray falls upon an uneven surface, the light is reflected in different directions. This image is then registered by a camera and analyzed with the use of a dedicated computer program. The Moire phenomenon is a result of interaction between two periodic structures [18,19]. It has been widely used in a number of studies evaluating posture in children and adults [20,21].An important advantage of this method is that photogrammetric evaluation is a non-invasive test which can be repeated many times without putting patient’s health at risk [22,23].This technique has also been used in previous studies to evaluate the posture of women treated for breast cancer [7,8,11,12].

The main advantages of this approach are that they allow for an evaluation of the body using the same image [24,25].

The lack of statistically significant differences between the groups proves that breast surgery has a greater impact on posture compared to axillary intervention itself [17]. The study showed that BCT+SLNB is associated with a lower rate of late complications.

The analysis of the side-dependent parameters proved that patients tend to tilt towards the operated side (TTA), raise shoulder on the operated side (SLA), inferior angle of the scapula tends to be raised on the operated side, and also the maximal lateral deviation of spinal processes can be observed on the operated side (VP). Additionally, the comparison of the side-dependent parameters showed that BCT+ALND patients raised their shoulder and inferior angle of the scapula significantly more often on the operated side.

The study showed that in BCT patients, thoracic kyphosis was more advanced, as expressed by increased GAMMA and BETA angles and a decreased TKA. The differences were obvious when compared to the control group.

The reason behind deepening kyphosis is both surgery and adjuvant treatment. As a result, the patients start to lead a more sedentary lifestyle, causing weakening of torso muscles and bone deformities [26]. Abnormal curvature of the spine results in a diminished ability to transfer the load and poor amortization [27]. Other studies suggest that, in breast cancer patients, physical activity is limited and the percentage of muscle tissue is lower and it causes the body postural changes [28,29].Another reason for increased kyphosis in patients after BCT may be the fact that these patients suffered from breast tissue defect. Patients who undergo BCT do not receive breast prostheses, which may further affect the development of deformations within the spine, especially in patients with large breasts and significant tissue defects. The studies conducted by Findikcioglu et al. revealed statistically significant differences in the angles of thoracic kyphosis and lumbar lordosis observed in patients after mastectomy with breast cup A and D [30].

Another factor which could influence postural imbalance may be post-operative radiotherapy. Previous research demonstrated that patients who received radiotherapy also reported greater problems with shoulder mobility and higher incidence of lymphedema or sensory disorders [31]. These aspects may lead to the development of postural disorders, adopting involuntary posture or forward bending of the trunk [32,33]. A similar effect can occur in patients for whom radiotherapy was applied to the breast and axillary fossa. The majority of patients included in our study received radiotherapy.

Post-operative radiotherapy, as well as the presence of scar within the breast and axillary fossa, may cause the retraction of cervical muscles and spasms in the trapezium and scalene muscles. Moreover, scars left after surgical treatment or radiation therapy result in the reduction of elastic properties and fibrosis of the skin [34].

Limited physical activity and fatigue are the main causes of weakening of muscles responsible for maintaining posture [35,36].The results were significantly worse in patients after axillary lymph node resection compared to the control group with respect to the following parameters: GAMMA, TKA, SLA (in mm) and SP (in mm).

It has been found that BCT+ALND patients showed stronger association of the operated side and higher position of the shoulder and scapula compared to BCT+SLNB patients. It can be explained by the fact that physical activity in BCT+ALND patients within the upper girdle is limited compared to patients without axillary lymphadenectomy [37]. Limited motion range of the shoulder girdle affects BCT patients regardless of the type of surgical intervention. On the operated side, the upper extremity is characterized by limited movement and motion range as well as muscle weakness, which leads to biomechanical changes and asymmetry between left and right side [38,39].

In our study, a statistically significant correlation between the operated side and torso tilting was observed. The patients tended to tilt to the operated side and raise the shoulder and inferior angle of the scapula on the operated side. Raised shoulder and scapula, lateral inclination towards the operated side—those are typical symptoms of so-called “half-woman complex”. Our study confirmed that it affects not only patients after radical surgery, but those who underwentto breast-conserving therapy as well [40,41].

Postural imbalance stems from both biomechanics and psychological disturbances. BCT patients require psychological support, which can help improve posture and has been emphasized by other authors [42]. It can be concluded that non-usage of breast prosthetics filling the tissue defect negatively influences posture after breast-conserving treatment. Depending on the breast size, the difference in load can be substantial [30,43]. However, not only surgical treatment affects posture. Other authors highlight the impact of hormonal therapy on bone density [44,45,46].

An important strength of this study is the evaluation of posture in women operated on due to breast cancer, which not only includes the type of surgical intervention in the axillary fossa, but also examines the control group. In our study, all patients from the experimental group received treatment in the same facility, which ensured reliability of medical data (the stage of clinical advancement of neoplastic disease, the types of complementary therapies, the number of removed lymph nodes and the type of surgical procedure). In our study, also the control group was carefully selected.

Our study, despite being the first report evaluating the influence of axillary intervention on posture, has several limitations. Despite adequate selection of patients to each group, our research is not a prospective study; evaluating the initial posture of patients before surgical procedure could provide a valuable insight to this research. Results from our study are creating a hypothesis for a prospective study in the future. Also, other confounding factors were not included, such as level of physical activity and participation in rehabilitation programs. Another limitation of our study is the fact that we have not analyzed the relation between postural changes and post-operative radiotherapy in the axillary fossa region in the group of patients who were operated on with the BCT+SLNB approach.

Lastly, there was no comparison according to hand dominance and the postural changes.

## 5. Conclusions

In our study, we demonstrated a lack of any statistically significant impact of type of axillary intervention (ALND vs. SLNB) on postural imbalance in breast cancer patients subjected to breast-conserving therapy (BCT). Our study showed that body posture in BCT patients was different from that of healthy women. It was manifested by deepened thoracic kyphosis, unequal shoulder levels, scapula protrusion. The abnormalities were more pronounced in patients after axillary lymphadenectomy.

## Figures and Tables

**Figure 1 jcm-10-01432-f001:**
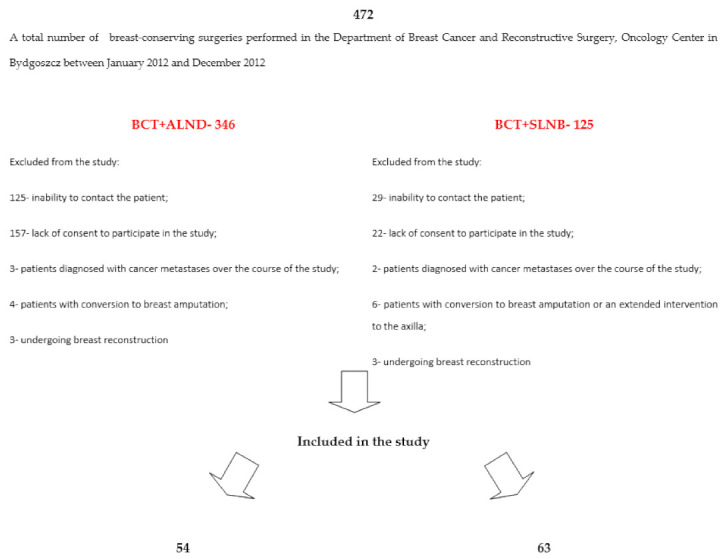
Study scheme describing patient exclusion from the study. BCT+SLNB: breast-conserving therapy+sentinel lymph node biopsy, BCT+ALND: breast-conserving therapy+axillary lymph node dissection.

**Table 1 jcm-10-01432-t001:** Sociodemographic and clinical data in studied group.

Variable	BCT+ALND (*n* = 54)	BCT+SLND (*n* = 63)	CG (*n* = 54)	Kruskal–Wallis Test /Chi^2/ Mann–Whitney’s U-Test	Dunn’s PostHoc Test
Age	S.D. = 60.04 Me = 61	S.D. = 60.67 Me = 62	S.D. = 59.76 Me = 62	*p* = 0.9877	1 vs. 2 *p* = 1.0000 1 vs. 3 *p* = 1.0000 2 vs. 3 *p* = 1.0000
Body weight	S.D = 69.57 Me = 66	S.D. = 72.94 Me = 73	S.D. = 74.37 Me = 72	*p* = 0.1179	1 vs. 2 *p* = 0.2626 1 vs. 3 *p* = 0.1748 2 vs. 3 *p* = 1.0000
Height	S.D. = 1.64 Me = 1.64	S.D. = 1.63 Me = 1.64	S.D. = 1.62 Me = 1.61	*p* = 0.1382	1 vs. 2 *p* = 1.0000 1 vs. 3 *p* = 0.1571 2 vs. 3 *p* = 0.5346
Operated side				*p* = 0.1929	
R	21 (38.89%)	33 (52.38%)			
L	33 (61.11%)	30 (47.62%)			
Clinical stage				*p* = 0.0098	
I A	30 (55.56%)	50 (79.35%)			
II A	21 (38.89%)	13 (20.65%)			
II B	3 (5.56%)	0 (0.0%)			
Number of dissected nodes	S.D. = 15.74 Me = 14.00	S.D. = 2.48 Me = 3		*p* < 0.0001	
Number of affected nodes	S.D. = 2.35 Me = 2	S.D. = 0.03 Me = 0		*p* < 0.0001	
Suplementary treatment CHTH, RTG RTG	45(81.48%) 9 (16.67%)	61 (96.83%) 1 (3.17%)		*p* < 0.0001	

*n*—number of patients, S.D.—standard deviation, Me—median, BMI—body mass index, L—left. R—right, BCT+SLNB—breast-conserving therapy+sentinel lymph node biopsy, BCT+ALND—breast-conserving therapy+axillary lymph node dissection, CG—control group, ER—estrogen receptor, PR—progesterone receptor, HER2—human epidermal growth factor receptor 2, RTH—radiotherapy, CHTH—chemotherapy, *p*—calculated probability value.

**Table 2 jcm-10-01432-t002:** Analysis of ALPHA, BETA, GAMMA, TKA, LLA, PTA (mm), SLA (mm) and SP (mm) in studied groups (BCT+ALND, BCT+SLNB, CG).

Postural Parameters	BCT+ALND (*n* = 54)	BCT+SLND (*n* = 63)	CG (*n* = 54)	Kruskal–Wallis Test	Dunn’s PostHoc Test
ALPHA	S.D. = 13.22 Me = 13.25	S.D. = 11.14 Me = 12	S.D. = 12.29 Me = 12.4	*p* = 0.1568	1 vs. 2 *p* = 0.1630 1 vs. 3 *p* = 0.8636 2 vs. 3 *p* = 1.0000
BETA	S.D. = 11.23 Me = 10.6	S.D. = 12.40 Me = 12.3	S.D. = 8.98 Me = 9.1	*p* = 0.002	1 vs. 2 *p* = 0.7587 1 vs. 3 *p* = 0.0796 2 vs. 3 *p* = 0.0015
GAMMA	S.D. = 13.53 Me = 13.65	S.D. = 13.31 Me = 12.3	S.D. = 11.18 Me = 11.3	*p* = 0.0044	1 vs. 2 *p* = 1.0000 1 vs. 3 *p* = 0.0047 2 vs. 3 *p* = 0.0547
TKA	S.D. = 151.75 Me = 154.85	S.D. = 154.29 Me = 154.8	S.D. = 159.85 Me = 160	*p* < 0.0001	1 vs. 2 *p* = 1.0000 1 vs. 3 *p* = 0.0001 2 vs. 3 *p* = 0.0001
LLA	S.D. = 155.55 Me = 155.95	S.D. = 156.46 Me = 158.1	S.D. = 158.74 Me = 158	*p* = 0.1212	1 vs. 2 *p* = 0.8961 1 vs. 3 *p* = 0.1200 2 vs. 3 *p* = 0.8524
PTA	S.D. = 0.99 Me = 0	S.D. = 1.59 Me = 0	S.D. = 1.09 Me = 0	*p* = 0.4869	1 vs. 2 *p* = 0.8792 1 vs. 3 *p* = 0.8847 2 vs. 3 *p* = 1.0000
SLA	S.D. = 6.03 Me = 5.8	S.D. = 5.76 Me = 5.8	S.D. = 4.15 Me = 2.9	*p* = 0.0004	1 vs. 2 *p* = 1.0000 1 vs. 3 *p* = 0.0008 2 vs. 3 *p* = 0.0040
SP	S.D. = 7.34 Me = 7.15	S.D. = 5.90 Me = 5.1	S.D. = 3.84 Me = 3.1	*p* = 0.0002	1 vs. 2 *p* = 0.1280 1 vs. 3 *p* = 0.0001 2 vs. 3 *p* = 0.0825

M—arithmetic mean. Me—median, BCT+SLND—breast-conserving therapy+sentinel lymph node biopsy, BCT+ALND—breast-conserving therapy+axillary lymph node dissection, CG—control group, *n*—number of patients, S.D.—standard deviation, Me—median, ALPHA—lumbosacral spine angle, BETA—thoracolumbar spine angle, GAMMA—thoracic spine angle, TKA—thoracic kyphosis angle, LLA—lumbar lordosis angle, PTA—pelvic tilt angle, SLA—shoulder level angle, SL—scapula level, mm—millimeters. ALPHA, BETA, GAMMA, TKA, LLA parameters are expressed in degrees, PTA, SLA, SP are expressed in millimeters, *p*—calculated probability value.

**Table 3 jcm-10-01432-t003:** Analysis of TBA, TTA, PTA, SLA, SL and VP in studied groups; analysis of correlations between groups and parameters in relation to the operated side (left or right).

Postural Parameters		L	R	chi^2 Test
TBA	Front	30 (55.56%)	34 (53.97%)	*p* = 0.9762
	Back	23 (42.59%)	28 (44.44%)	
	Zero	1 (1.85%)	1 (1.59%)	
TTA	Left	36 (66.67%)	28 (44.44%)	*p* = 0.0264
	Right	18 (33.33%)	32 (50.79%)	
	Zero	0 (0.00%)	3 (4.76%)	
PTA	Left	18 (33.33%)	15 (23.81%)	*p* = 0.3811
	Right	4 (7.41%)	3 (4.76%)	
	Zero	32 (59.26%)	45 (71.43%)	
SLA	Left	36 (66.67%)	7 (11.11%)	*p* < 0.0001
	Right	18 (33.33%)	56 (88.89%)	
	Zero	0 (0.00%)	0 (0.00%)	
SL	Left	30 (55.56%)	16 (25.40%)	*p* = 0.0010
	Right	8 (14.81%)	26 (41.27%)	
	Zero	16 (29.63%)	21 (33.33%)	
VP	Minus	39 (72.22%)	6 (9.52%)	*p* < 0.0001
	Plus	15 (27.78%)	57 (90.48%)	
	Zero	0 (0.00%)	0 (0.00%)	

M—arithmetic mean, Me—median, BCT+SLND—breast-conserving therapy+sentinel lymph node biopsy. BCT+ALND breast-conserving therapy +axillary lymph node dissection. L—left. R—right. CG—control group. TBA—torso bend angle. TTA—torso tilt angle. PTA—pelvic tilt angle. SLA—shoulder level angle. SL—scapula level. VP—vertebra position. TBA, TTA, PTA, SLA, SL parameters are expressed in degrees. VP parameter is expressed in millimeters. *p*—calculated probability value.

**Table 4 jcm-10-01432-t004:** Analysis of side-dependent (left or right) parameters in BCT+ALND and BCT+SLNB groups; correlations between groups (BCT+ALND vs. BCT+SLNB).

Postural Parameters		BCT+ALNB		BCT+SLNB		Homogeneity
		L	P	L	P	
TBA	Front	13 (61.95%)	17 (51.51%)	17 (53.12%)	17 (58.62%)	0.3986
	Back	8 (38.05%)	16 (48.49%)	15 (46.88%)	12 (41.38%)	
TTA	Left	16 (76.19%)	16 (53.33%)	20 (60.61%)	12 (40.00%)	0.8121
	Right	5 (23.81%)	14 (46.67%)	13 (39.39%)	18 (60.00%)	
PTA	Left	7 (87.50%)	8 (72.73%)	11 (78.57%)	7 (100%)	0.1229
	Right	1 (12.50%)	3 (27.27%)	3 (21.43%)	0 (0%)	
SLA	Left	18 (85.71%)	1 (3.03%)	18 (54.55%)	6 (20.00%)	0.0028
	Right	3 (14.29%)	32 (96.97%)	15 (45.45%)	24 (80.00%)	
SL	Left	16 (84.21%)	5 (17.86%)	14 (73.68%)	11 (78.57%)	0.0018
	right	3 (15.79%)	23 (82.14%)	5 (26.32%)	3 (21.43%)	
VP	Minus	16 (76.19%)	1 (3.03%)	23 (69.70%)	5 (16.67%)	0.0800
	Plus	5 (23.81%)	32 (96.97%)	10 (30.30%)	25 (83.33%)	

M—arithmetic mean, Me—median, BCT+SLND—breast-conserving therapy+sentinel lymph node biopsy, BCT+ALND—breast-conserving therapy+axillary lymph node dissection, L—left, R—right, CG—control group, ALPHA—lumbosacral inclination angle, BETA—thoracolumbar inclination angle, GAMMA—thoracic inclination angle, TBA—torso bend angle, TTA—torso tilt angle, PTA—pelvic tilt angle, SLA—shoulder level angle, SL—scapula level, VP—vertebra position, TBA, TTA, PTA, SLA, SL, VP parameters are expressed in degrees, *p*—calculated probability value.

## Data Availability

The datasets generated and/or analyzed during the current study are available from the corresponding author on responsible request.

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
