# Peer review of "Evaluation of Late Postural Complications in Breast Cancer Patients Undergoing Breast-Conserving Therapy in Relation to the Type of Axillary Intervention-Cross-Sectional Study"

_jcm, 2021, doi:10.3390/jcm10071432_

Round 1

Reviewer 1 Report

I would appreciate your submission of your manuscript, titled `Evaluation of late postural complications in breast cancer patients undergoing Breast Conserving Therapy in relation to the type of axillary intervention-cross sectional study”

This manuscript is an interesting topic, but it needs some confirmation and explanation.

First, it has been widely known that postural changes in breast cancer survivor mainly occur after mastectomy rather than Breast conserving surgery(BCS). A further review of the literature on postural changes after BCS should be presented in introduction and discussion. And, the differences of postural changes between mastectomy and BCS should be mentioned.

Second, the reasons for deepening kyphosis in BCS patients compared to control was explained by the sedentary lifestyle and limited physical activities, but the evidence is insufficient. We cannot agree that BCS patients have limitations in their daily activities or occupational life. Another reasons or mechanism is needed to explain this results.

Third, the reliability of photogrammetric evaluation using Moire phenomenon needs to be added in the discussion.

Author Response

Dear Reviewer,

Thank you for reviewing our article titled ‘Evaluation of late postural complications in breast cancer patients undergoing Breast Conserving Therapy in relation to the type of axillary intervention- cross-sectional study’ We deeply appreciate your opinion as well as constructive comments that contributed to more profound consideration of issues addressed in our publication. The comments in your review will guide us in our future work.

            In response to those comments we:

- presented the differences in postural changes between the mastectomy and BCS groups in the Introduction and Discussion section.

- provided for more details on the reasons for increased kyphosis in BCT patients when compared to the control groupin the Discussion section. We included for examplesuch factors as post-operative scar and implementation of post-operative radiation therapy.

- we described in the Discussion section the reliability of photogrammetric evaluation with the use of Moiré phenomenon.

Thanks to the comments received, we were able to refine the publication in terms of the content. All errors mentioned in the review were corrected in the final version of the publication.

                                                                                                          Kind regards,

                                                                                                          the Authors

Reviewer 2 Report

A very well-conducted study: it is well written, intelligible, consistent and formally impeccable. It explored an issue that is underestimated and could be prevented by an early rehabilitation programme.

There are only a few aspects that to be clarified

1- You never mentioned in the text the irradiation which is obvious patients were submitted to (due to its complementarity in BCT). To be exhaustive you should specify the dose and the scheme of radiotherapy they received; secondly, if they received RT as well as on the breast also on other regions such as the axilla for example (because it could represent a bias in terms of long term effects on the issue you are examining).

2- I noticed, as you described in Fig.1, that you had a very high number of ALND if compared with SLNB: on 472 pts, 364 were treated with BCT+ALND and 125 with BCT+SLNB. Could you explain why?

Author Response

Dear Reviewer,

Thank you for reviewing our article titled ‘Evaluation of late postural complications in breast cancer patients undergoing Breast Conserving Therapy in relation to the type of axillary intervention- cross-sectional study’. We deeply appreciate your opinion as well as constructive comments that contributed to more profound consideration of issues addressed in our publication. The comments in your review will guide us in our future work.

                        In response to those commentaries, we clarified the Materials and Methods section where we described radiotherapy treatment regimens provided to our patients. We agree with you that patients who were treated with BCT approach, could have received radiation to the axillary fossa region, which could in turn influence their posture. We brought up this aspect in our discussion and included it in the limitations section of our study.

There wasan error in Fig.1 and we reanalyzed this data. As many as 125 patients were treated with BCT+ALND, while 346 patients received BCT+SLND therapy. We have corrected this information and would like to express our gratitude for your remark once more.

Thanks to the comments received, we were able to refine the publication in terms of the content. All errors mentioned in the review were corrected in the final version of the publication.

                                                                                                          Kind regards,

                                                                                                          the Authors

Round 2

Reviewer 1 Report

Dear author,

The previously reviewed parts have been well revised and added with better contents.

Thank you.